# Soil Fertility Clock—Crop Rotation as a Paradigm in Nitrogen Fertilizer Productivity Control

**DOI:** 10.3390/plants11212841

**Published:** 2022-10-25

**Authors:** Witold Grzebisz, Jean Diatta, Przemysław Barłóg, Maria Biber, Jarosław Potarzycki, Remigiusz Łukowiak, Katarzyna Przygocka-Cyna, Witold Szczepaniak

**Affiliations:** Department of Agricultural Chemistry and Environmental Biogeochemistry, Poznan University of Life Sciences, Wojska Polskiego 28, 60-637 Poznan, Poland.

**Keywords:** nitrogen, nitrate-nitrogen, nitrogen use efficiency, nitrogen-supporting nutrients, phosphorus, potassium, maximum attainable yield, soil fertility management

## Abstract

The Soil Fertility Clock (SFC) concept is based on the assumption that the critical content (range) of essential nutrients in the soil is adapted to the requirements of the most sensitive plant in the cropping sequence (CS). This provides a key way to effectively control the productivity of fertilizer nitrogen (N_f_). The production goals of a farm are set for the maximum crop yield, which is defined by the environmental conditions of the production process. This target can be achieved, provided that the efficiency of N_f_ approaches 1.0. Nitrogen (in fact, nitrate) is the determining yield-forming factor, but only when it is balanced with the supply of other nutrients (nitrogen-supporting nutrients; N-SNs). The condition for achieving this level of N_f_ efficiency is the effectiveness of other production factors, including N-SNs, which should be set at ≤1.0. A key source of N-SNs for a plant is the soil zone occupied by the roots. N-SNs should be applied in order to restore their content in the topsoil to the level required by the most sensitive crop in a given CS. Other plants in the CS provide the timeframe for active controlling the distance of the N-SNs from their critical range.

## 1. Introduction—The Battle for Yield

The required increase in total food production of around 70% in 2050, compared to 2010, will largely depend on increases in the yields of crop plants, which serve as staple food for humans [1,2]. At present, meeting the demand for food in the next 28 years, in connection with the required reduction in greenhouse gas emissions, poses a major challenge for global organizations (e.g., the United Nations), the Food and Agriculture Organization (FAO), and national governments [3,4]. Russia’s war on Ukraine has clearly highlighted the importance of net food producers, such as Ukraine, in stabilizing the global food market. In the 2020/21 season, this country exported 69.8 million tons of all cereals, accounting for 11.8% of the global export. Ukraine’s share in global exports of sunflower seeds and oil exceeded 50% (52%). The war collapsed exports from Ukraine, not only of cereals and sunflower products, but also soybeans and rapeseed [5,6]. Strubenhoff [6] has indicated four groups of activities that can be urgently taken by world leaders to save the world from hunger. Concerning the national level, he pointed to the need to change the food policies of the EU and the USA. In the case of the USA, the author suggested reducing the production of biofuels. In the case of the EU, the author indicated the need to move away from policies on reducing the use of mineral fertilizers. The general conclusion formulated by Strubenhoff [6] was as follows: “For the time being, we need more production, not less. Climate objectives are good to save the planet, but we also need to feed the people on the planet*”*. This should be seen as a motto for political and environmental players who truly understand the functioning of Planet Earth in a holistic, not reductionist sense. 

The increase in food production by 2050 will in fact be the result of two main drivers, including the increases in arable land area and yields of main staple crops. The first, the main factor in covering the food gap by 2050—is in fact a key constraint, as evidenced by the wide discourse pursued by decision makers in the area of food policy. This goal, in fact, is limited by a lack of fertile soils. The potential resources for expanding arable land area are predominantly in the tropics. However, these soils, despite high natural fertility, require large inputs of the means of production [7,8]. Moreover, the destruction of the rain forest is expected to completely disrupt the Earth’s climate [9]. Thus, in order to cover the food gap by 2050, the main challenge facing the world is to increase the yields of crop plants in “old agricultural areas” [10]. The share of the second—but dominant—factor in covering the food gap by 2050 (crop yields), has been estimated at over 80% [11,12]. There are four main factors that are considered crucial in actions oriented toward yield increases. The first is breeding progress. Meeting the 2050 target requires an annual increase in the yield of major crops such as wheat, maize, rice, and soybean at the level of 2.4% annually. This target, as reported by Ray et al. [13], will not easily be met, as the current yield increases of these crops are much below this target, reaching in relative terms only 67% for wheat, 42% for maize, 38% for rice, and 55% for soybean. The second factor is inherent in the effective use of mineral fertilizers and other crop protection measures. The actions taken by farmers should, however, be in line with the main assumptions of the concept known as Intensification of Sustainable Agriculture, which indicates the effective use of production means, including fertilizers [14]. The term “battle for yield,” as proposed in the title of this chapter, in the current geopolitical context refers precisely to the productivity of the basic unit of plant production (i.e., a single field) which, in fact, defines the homogenous fertility unit of the field [15]. 

The third factor—and, indeed, a dominant factor in the food production sector—is the effective use of fertilizer nitrogen (N_f_). The N_f_ consumption, as forecast for 2050, is 76% higher than in 2000 (181 vs. 103 mln t y^−1^) [4]. In another study, the increase in the demand for N_f_ in 2050, compared to 2005, will be in the range of 43–73% [16]. The crucial problem with N_f_ use by farmers—both for production and, consequently, for the environment—is its low efficiency (recovery). Limiting the effectiveness of N_f_ to the right N dose, fertilizer approach, and even the timing of its application is a dramatic simplification of a complex problem [17,18]. A reduction in N_f_ consumption, in light of the drastic increase in N fertilizer prices in 2022, seems more realistic at present [19]. However, the sudden drop in N_f_ consumption, as observed in 2022 in Europe, could disrupt the global food production chain. The maintenance of the level of N_f_ consumption is crucial for food production in the old agricultural areas in the world [20]; for example, a simulation regarding reduced N_f_ consumption in the U.S. for maize and rice indicates a possible decrease in yields of 41% and 27%, respectively [21]. These theoretical considerations from 10 years ago should be considered, in the face of the current fertilizer crisis [6,19].

The fourth factor, which is decisive for the efficient use of production measures in agriculture, is the knowledge and skills of the farmers and their advisers, which are necessary to exploit the yield potential of the currently grown varieties. The real challenge for the farmer in the effective use of N_f_ is the correct diagnosis of the plant demand for N in the most critical phase(s) of the yield formation. At this particular stage of crop plant growth, a synchronization between the plant’s requirements for N with its supply from soil (soil N resources plus controlled N_f_ application) is crucial for the formation of yield components. The farmer must know and recognize both the phases in which the plant builds up the yield components and the phases in which they are reduced. The importance of this issue can be presented through the examples of three crops. The first example concerns cereals, which deliver about 60% of carbohydrates and 50% of proteins to the world food market [22]. The key yield component determining the grain yield is the number of grains per unit area (grain density; GD). The most critical period of GD formation extends from the heading phase through flowering to the early milk (BBCH 71) [23,24]. Therefore, it can be called “the critical cereals’ window”. The second primary component of the yield is the grain weight (1000 grain weight; TGW), which is established during the grain filling period (GFP). This period extends from the early milk stage (BBCH 72) to plant maturity (BBCH 90) [25]. Its impact on grain yield is much lower than that of GD [26]. For the second example, maize is a crop producing the greatest amount of food for humans or fodder for livestock [27]. The critical period of the primary yield’s component formation is the stage of fifth leaf, in which the cob initials are formed. The key nutrient responsible for this process is the supply of N [28,29]. The third example is potatoes, the importance of which as food for humans has been growing rapidly [30]. The critical period for tuber yield establishment is tuberization [31]. The tuber yield depends on the number and weight of young tubers. These processes are driven by the supply of N, but also require a good supply of potassium (K) and phosphorus (P), at least [32,33]. These three examples allow us to conclude that the farmer’s knowledge about the functions of N in plants is the absolute basis for determining an effective technological solution for the cultivated crops. 

The basic questions to be asked here are as follows:(1)Is the increase in nitrogen use efficiency (NUE) the real challenge for the increase in yields?(2)How is the effect of nitrogen-supporting nutrients (N–SNs) on efficiency of N_f_ manifested?(3)What is the required level of N–SNs in the soil, in order to maximize the N_f_ yield-forming effect?

The third question is essentially sets the goal for this conceptual article. 

## 2. A New Paradigm of Nitrogen Use Efficiency Control—The Basis of the Concept 

There are five general assumptions to consider before any discussion or forecast of crop production outcomes (i.e., the yields) by scientists, farmers’ advisors, and food policy makers. First, the amount of solar energy reaching any part of the Earth’s land surface is determined by its geographical location [34]. In agriculture, the basic unit of analysis is a single field or its homogenous production part, which is directly managed by the farmer [15,35]. Second, the accumulation of dry matter by a plant during its life cycle, its growth rate, dry matter partitioning between plant organs, and subsequent remobilization depend on the water and N supply. Third, the amount of available N in the plant rooting zone during the growing season is critical for the formation of yield components and, consequently, for the yield. Fourth, the uptake and use of N by the plant depends on the availability of other essential nutrients present in the plant’s rooting zone. Fifth, the capacity of the soil and its potential to provide these nutrients to plants is limited. Their content in the plant natural growth milieu (that is, soil) is not infinite and, therefore, needs to be both controlled and supplemented by the farmer.

The effects of climatic and soil factors (environmental conditions) on crop growth and yield are manifested in terms of the maximum attainable yield (Y_attmax_), which can be reached in well-defined geographic locations [36,37]. It is necessary to take into account that the use and impact of non-nutritional production factors on the grown crop is the result of a farmer’s decision and/or legal limitations. Thus, on a specific field on a farm, the main issue for the farmer is to consider the effective management of N, which, in fact, depends in the current state of soil fertility. Sustainable management of N in the field should, therefore, be considered as a balance between necessary and sufficient conditions:(1)The necessary condition is the actual yield, which is a function of the amount of available N in the rooting zone of the currently grown crop.(2)The sufficient condition is the yield-forming functions of nutrients other than N to support the action of N by increasing its uptake and its use by the currently grown plant.

In fact, the necessary condition concerns the control of efficiency, which is the productivity of available N present in the soil zone occupied by plant roots during the growing season. The more detailed assumptions are as follows:(1)A crop plant in a well-defined geographic area, provided stable environmental and nutritional conditions, can reach Y_attmax_.(2)The key production factor is N, present in the soil or/and supplied to the plant as fertilizer (natural, manure; mineral, N_f_).(3)All other nutrients, called nitrogen supporting nutrients (N-SNs), affect the Y_attmax_, in relation to their relative deficiency in available form in the plant rooting zone.

These relationships can be expressed as a set of general formulae:Actual yield:
(1)$$Y_{a}={\;Y}_{\mathrm{attmax}}\times \mathrm{EN}$$
2.Nitrogen Efficiency (EN):
(2)EN=EP ×EK ×EMg ×ES ×… …× ETi where Y_a_ denotes the actual yield; Y_attmax_ denotes the maximum attainable yield; EN is the fractional value of NUE; and EP, EK, …, ETi are the fractional efficiencies of various N-supporting nutrients (N-SNs).

When the fractional value of an N-SN’s efficiency index approaches 1.0 (≤1.0), it indicates a sufficient range of content of the N-SN in available form in the soil. However, any deviation from 1.0 indicates a disturbance in the supply of a given nutrient to the currently grown crop, thereby reducing the NUE. Summarizing the above assumptions, it should be clearly stated that the key challenge for the farmer is to achieve high productivity of N fertilizer (N_f_), but without a reduction in yield. This goal is achievable, provided that the critical level (sufficiency range) of soil fertility is achieved for N–SNs. Only at the equilibrium state between the supplies of N and N–SNs to the plant during the growing season—in particular, in the critical phases of yield component formation—is it possible to effectively exploit the yield potential of the currently cultivated plant.

## 3. Nitrogen—A Unique and Critical Factor in Plant Production

There exists a general consensus that the choice of a cultivar with well-defined yield potential is the basis of crop cultivation. Exploitation of the crop potential, however, essentially depends on the supply of water and N. For these reasons, these production factors are defined as yield-limiting [38,39]. These factors cannot be, however, treated as substitutes [40]. Water is a growth factor that regulates the plant temperature and, as a consequence, its whole metabolism and growth [41]. It has been well-documented that the amount of water available to the plant during the growing season is the result of both the water retention capacity of the soil and current precipitation. These two factors determine the Y_attmax_ under well-defined climate and soil conditions [36]. Water acts as a natural carrier of nutrients, both in soils and in the plant [42].

The plant is an autotrophic organism which, in order to close its life cycle, must be supplied with adequate amounts of both water and nutrients at well-defined stages of growth [43]. Plant growth can be defined as a set of processes in which both the plant and the soil—as its natural growth medium—interact with each other throughout the growing season. The importance of N for plant growth and yield results from its presence in key biological molecules [42]. The key N-dependent enzyme, which is decisive in the survival of life on Earth, is the ribulose bisphosphate carboxylase-oxygenase enzyme, simply called Rubisco (RuBP). Its key function is the capture and subsequent fixation of the CO_2_ molecule, which is the basic substrate for the production of elementary sugar compounds [44,45]. The total mass of Rubisco in terrestrial plants has been estimated at ≈0.7 Gt. This enzyme constitutes 2.5–3% of the total leaf weight of leaves and about 50% of the total leaf proteins [46]. A hypothesis has recently emerged that Rubisco may also be treated a source of N during protein synthesis. This phenomenon is revealed only under conditions of excessive CO_2_ capture by the plant in the circadian cycle [46].

N, mainly as nitrate (N-NO_3_), also acts a local and systemic signaling molecule involved in the current regulation of the hormonal status and morphology of the plant [47,48]. For this reason, this inorganic N form has recently been termed a plant morphogen [49]. Clear evidence for the dominant role of N-NO_3_ in yield formation is its influence on plant growth, which affects both the shoot and root system architecture [50]. The effect of N supply to the plant manifests itself in clear, visible changes in the architecture of the plant’s canopy. As can be seen in Figure 1a, wheat plants grown on an N control plot (i.e., without N_f_ supply) presented stunted growth (i.e., dwarf stature), low weight and surface area of leaves, and pale green color. In contrast, plants well-fed with N were characterized by a well-developed shoot, high mass and surface area of leaves, and an intense green color. All of these plants, despite a significant difference in the architecture of shoots, were in the same phase of growth (i.e., booting; BBCH 40–49). This phase is the crucial for the development of yield structure and determines the number of fertile florets [51]. Excess N supply to the plant, as shown for maize, results in the establishment of more cobs per plant (Figure 1b). However, this does not mean a higher yield of grain. Excessive supply of N also results in excessive growth of non-productive plant parts, leading to a reduction in grain per unit area [52,53].

The introduction of new cereal phenotypes in the 1960s, such as varieties with reduced stem length, first for wheat and then for rice, significantly changed the shoot architecture (dwarfism of the shoot). These genetic modifications resulted in an increased harvest index (HI)—that is, the share of grain in the total shoot biomass—at the expense of the stem. Improvement of the harvest index (HI) is the greatest effect of the Green revolution, as it finally led to higher grain yields of cereals, including rice [54]. However, the exposition of dwarf genes has also caused a reduction in the root system size of wheat varieties, which is significantly smaller than that of the classic ones [55,56]. As a consequence, the semi-dwarf or dwarf growth mode of modern cereals varieties result in the higher yield, provided that the supply of nutrients (especially N_f_) is high, and that the plants are strongly protected against pathogens [54,57]. It can, therefore, be concluded that the currently grown cereals, due to their high requirements for N on one hand, and their smaller root systems on the other hand, are extremely sensitive to the supply of nutrients responsible for the uptake of N from the soil. One of the proposed solutions aimed at the increase in nitrogen use efficiency (NUE) are new-generation varieties that are capable of developing deep root systems. The proposed ideotype of this root system—referred to as “steep, cheap, and deep”—assumes the effective uptake of water and dissolved nutrients (mainly nitrates) [58,59]. A reorientation of the current breeding approach is urgently required in intensive production systems, where high rates of N_f_ are typically applied. It can also provide a good solution in areas with frequent periods of drought, regarding the main phases of plant growth.

## 4. Nitrogen-Supporting Nutrients

Plant growth and productivity are the result of the action of about 20 elements that must be present in the soil to complete the plant’s life cycle. The biophysical functions of plant-related elements have been well-documented and presented extensively in textbooks and review articles [60,61,62]. Not all of these elements are considered as nutrients, but all of them have a positive impact on the yield of crop plants [63]. A typical example is titanium (Ti), the positive effect of which on many crops has been recently documented [64].

N, considered especially in the form of nitrate (as discussed in the previous section), is the key nutrient, affecting both the rate of plant growth and the formation of yield components. The key evidence, in addition to that discussed above, is the response of the yield to the application of N–P–K fertilizers in various mutual fertilization systems. The effect of the interactions between N and other basic nutrients has been well-presented in long-term static fertilization experiments [65,66,67]. As shown in Figure 2, winter rye cultivated on Luvisol in a 7-course crop rotation (including two years of alfalfa) for 40 years yielded on plots without K (NP) or P (NK) only 6% and 5% less than on the NPK plot. The lack of both nutrients (i.e., K and P), as evidenced on a plot fertilized only with N, resulted in a yield drop of only 2.5%. The yield on the absolute control (AC) plot (i.e., the plot without application of any fertilizer, mineral or organic) for 40 years, was 30% lower than that on the NPK plot. Moreover, the same level of yield was recorded on plots fertilized only with P or K. Slightly greater differences between fertilization treatments were recorded for spring barley. Relative yield reductions were: −10%, −7%, and −42%, for NP, NK, and AC, respectively, as compared to NPK. The same yield level as for AC was noted on plots fertilized only with P or K. It must be added that the yields on the N plot were lower by 10%, compared to NPK [65]. The above-documented trends of plants grown on the naturally low fertility soil (Luvisol), with respect to different combinations of basic nutrients, have been supported by data from fertile soils, such as Entisol in the Netherlands (calcareous Entisol, containing 30% clay, 10% CaCO_3_. The obtained data indicated that a lack of K fertilization over a period of 28 years did not adversely affect the yield of four crops grown in four-course crop rotation (sugar beets, spring barley, potatoes, and winter wheat) over 28 years. The lack of P was much more important, as its lack reduced the yield of sugar beets by 11%, but those of potatoes and spring barley by only 7% [68]. These two examples clearly show that the main source of P and K for crops is soil.

All plant nutrients support plant growth and yield formation through their impact on the productivity of N which, to a great extent, depends on the application of N_f_
Figure 1a,b. It can be concluded that it is unrealistic to expect an increase in the yield of a modern variety, as discussed above, without delivering N from external sources, whether natural (e.g., manure) or mineral. For these reasons, all nutrients affecting plant growth and yield can be called “nitrogen-supporting nutrients” (N-SNs). Based on agronomic practice, the whole set of N-SNs can be divided into four groups: (i) basic macronutrients, such as potassium (K) and phosphorus (P); (ii) secondary macronutrients, including magnesium (Mg), sulfur (S), and calcium (Ca); (iii) micronutrients, such as iron (Fe), manganese (Mn), zinc (Zn), copper (Cu), boron (N), molybdenum (Mo), and chloride (Cl); and (iv) the beneficial group, composed of nickel (Ni), sodium (Na), silicon (Si), and titanium (Ti) [61,69].

The second important stage in the development of an economically and environmentally sound fertilization system using N_f_ requires knowledge regarding the patterns of accumulation of N-SNs during the growing season. The classic pattern for high-yielding winter oilseed rape (WOSR) is presented in Figure 3. There are some characteristics “hotpoints” in N-SN in-season patterns that require special attention by the farmer, based on the course of N accumulation. The most important points are:(1)The higher accumulation of K over N during the whole season of WOSR growth:
a.Starting with the rosette stage in Spring;Achieving the maximum K uptake over N at the full flowering stage (K_2_O:N as 1:1.6);Declining from flowering to maturity (K_2_O:N as 1:1).(2)Slow early growth in P uptake, continued up to the inflorescence stage (BBCH 50), followed by rapid ingrowth, lasting until the full flowering stage (BBCH 65), and then smoothly decreasing up to maturity.(3)A similar pattern for Mg as for P, but at a much lower level.(4)A spectacular pattern of Ca accumulation. Its uptake increases sharply at inflorescence, reaching a maximum at the end of the pod growth stage.

**Figure 3 plants-11-02841-f003:**
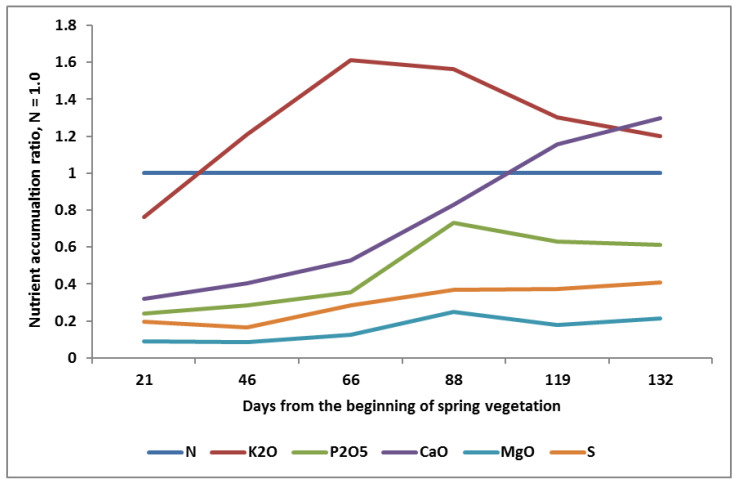
Patterns of nutrient accumulation during the growing season by high-yielding winter oilseed rape (simulation based on [70]).

The presented patterns of N, P, and K accumulation by WOSR during the growing season, obtained at the end of the 20th century, were very similar to their trends observed in the 1980s [71]. The same pattern has been observed for high-yielding rice [72]. The effective production of maize depends on the supply of K and N during the period preceding flowering, but requires stabilization of K accumulation up the milk stage [73]. It can be concluded that the high yield of the seed crops can be achieved, provided that the K:N ratio is higher than 1.0 during the vegetative phases for seed plant growth. In the light of the available data, the maximum K:N should be revealed for the high-yielding crop just before the end of the linear phase of its growth [74]. The seasonal pattern of basic nutrient accumulation by legumes is only slightly different. As observed for soybean, the maximum K was at the late vegetative stages, and the level then decreased smoothly, while the accumulation of N, P, Ca, Mg, and S progressed through to maturity. Moreover, the maximum K:N ratio was 0.6:1 [75].

## 5. Potassium

The biophysical functions of K are well recognized and described [76,77]. The most important functions of K in crop production are those that affect (i) nitrate-nitrogen uptake, (ii) protein synthesis, (iii) water management (i.e., control of the stomata circadian rhythm), (iv) growth of vegetative organs (i.e., transport of assimilates from leaves to the new buds and tissues), and (v) yield formation (i.e., transport of assimilates from leaves to growing fruits). All of these functions are inherent in the plant’s growth cycle [72,78,79].

The deficiency of K during the growing period of the cultivated plant adversely affects N uptake and, thus, photosynthesis, leading to a reduction in assimilate production [80]. K deficiency in the first stages of plant growth reduces the growth of single cells. As a consequence, the growth rate of new tissues and organs is reduced (Figure 4a); this applies to both roots and shoots. The reduction in shoot biomass is slightly lower than that in root biomass [80,81]. As documented for sugar beet, the growth rate of fibrous roots is much faster in soil rich in available K, compared to the soil with medium content [82]. The rate of plant cell expansion depends on the action of auxins, the concentration of which in the plant and transfer to the roots is strongly related to the availability of N-NO_3_ in the soil [49,83]. The stunted stature of crop plants is the most striking visual symptom of K deficiency (Figure 4). In maize, the length of internodes is drastically reduced during the phase of shoot intensive growth (Figure 4b). In cereals, the classical symptom is the same (i.e., reduced length of the stem). The secondary outcome, which is important in grain production, is the reduced density of ears (Figure 4a). This yield component will not be compensated for by a larger number of grains per ear or weight, thus directly leading to a significant decrease in grain density [24,84].

Crop plants differ in their demand for K. A proposed grouping of crop plants, according to K accumulated at harvest, and proposed recently by White et al. [85], has significant weaknesses. In general, it is true that cereals or legumes accumulate less K than dicotyledonous crops (leafy, root, or tuberous plants). It cannot, however, be concluded from this that cereals are tolerant of low soil K fertility level. It has been well-documented that cereals and cruciferous crops develop a large and intensive root system which, in turn, increases the absorption area of the plant for K uptake [86,87]. This specific plant feature determines the rate of K uptake under critical conditions, such as low content of available K or mild water stress. Sandy soils, as compared to loamy soils, as a rule are poorer in available K. Moreover, a decrease in the content of available water results in a much faster decrease in the coefficient of K effective diffusion [82,88,89]. The greater sensitivity of dicotyledonous plants to the level of soil K fertility is the result of two reasons [74,86]:(1)Higher demand for K in the linear phase of growth;(2)Much smaller root system, especially root length density.

Numerous scientific articles and even academic textbooks have presented—seemingly true—opinions regarding the quantitative dominance of N uptake over K accumulated by crops during the growing season [61,90]. This opinion can be applied to low-yielding crops, due to the deficiency of K in the linear phase of the plant growth [91]. In the case of high-yielding crops (i.e., those which are able to exploit the potential of cultivated varieties), the dominance of K uptake over N has been well-documented (Figure 3; [72,82]). The classic example is maize [53]. This crop takes more K than N during the vegetative growth. This opinion is also in accordance with a study on nutrient accumulation by wheat in India [92]. The authors clearly showed that the unit accumulation of K was higher than N, and the optimal N:P:K ratio in the plant dry matter for high-yielding wheat was as 6.6:1:8.1.

Based on the amounts of key nutrients accumulated by winter oilseed rape (WOSR), three regression models of the seed yield have been developed [93]. It is necessary to emphasize that the yield of WOSR ranged from 2.223 to 5.807 t ha^−1^. The obtained regression models were as follows:N → Y = 0.12N + 1.192          for n = 18, R^2^ = 0.69 and *p* ≤ 0.01, (3)
K → Y = 0.007K + 1.6          for n = 18, R^2^ = 0.89 and *p* ≤ 0.01, (4)
Mg → Y = −0.005 Mg^2^ + 0.39 Mg − 2.242     for n = 18, R^2^ = 0.61 and *p* ≤ 0.01 (5)

The final K uptake, regardless of the seed yield and weather conditions during the research, always exceeded that of N. The linear course for N and K clearly indicates that both nutrients were limiting factors for the yield. Moreover, the strength of the effect of N on WOSR yield, according to the R^2^ coefficient, was much weaker compared to that of K. This model of N and K accumulation by seed crops is not new, having been defined about 40 years ago for winter wheat and WOSR [71,94].

The seemingly contradictory opinion regarding the impact of N and K on yield is explained in Figure 5. The amount of K in winter wheat (WW) during the Yield Formation Period (defined by the linear increase in dry matter)—a mega-phase covering the main phases such as stem elongation, booting, and heading—exceeded, regardless of the water conditions, the amount of N taken up. Moreover, the K:N ratio up to the beginning of wheat flowering for irrigated wheat was higher than 1.0. A sharp drop in K accumulation was revealed during the grain filling period. The course of N was significantly different from that of K, showing a net increase in its accumulation up to the early milk stage (BBCH 72), then decreasing slightly. These two (partly opposing) trends resulted in the K:N ratio narrowing at wheat maturity. The same trend of K accumulation has been recently observed in barley [95]. The presented data clearly explain the discrepancy in published data regarding the trend of K accumulation by high-yielding crops.

The importance of K for yield formation is to be considered through the expression of yield components. It has been well-documented that K accumulation in seed crops reaches its maximum before flowering (Figure 3 and Figure 5). In seed crops, this period is crucial for establishing the seed/grain density, which is treated as the key yield component [96]. It should be clearly stated that an adequate K supply to the seed crop during the Yield Formation Period (YFP) supporting N supply is the prerequisite of high yield. The presented opinion is neither a hypothesis nor an assumption, but a conclusion from our own studies and supported by available literature sources. The excessive uptake of K by wheat may result, however, in GD reduction, consequently leading to a decrease in grain yield [97]. This phenomenon has also been observed in maize and dicotyledonous plants, such as potato [52,98,99,100]. The unexpected effect of excessive K uptake can be explained by the accelerated supply of N to plants due to co-transport of NO_3_^−^ and K^+^ ions through the plasma membrane [62,80].

The role of K in the transportation of assimilates in the phloem has been well-documented [60]. The survival of seeds/grain in the period from setting up to the watery stage is inherently related to the supply of assimilates [101]; therefore, it can be concluded that K is responsible for the final seed/grain set. This concept is supported by the impact of the accumulated K on yield components, as shown in Equation (1). As recorded by Grzebisz at al. [93], K uptake at the beginning of WOSR flowering was the key factor, positively affecting the seed density (SD), while negatively affecting Thousand Seed Weight (TSW) at the same time:SD = 203.3K + 3462 for n = 18, R^2^ = 0.92, *p* ≤ 0.01 (6)
TSW = −0.04K + 691 for n = 18, R^2^ = 0.48, *p* ≤ 0.05 (7)

The first equation fully corroborates the opinion of Pan et al. [102], who clearly stated that a deficiency of K reduces the sink size capacity. The second equation clearly confirms the phenomenon known as the dilution effect [103], concerning the dilution of nutrients in seeds, including N. Most important is the fact that the productivity of 1 kg of N_f_ increased from 14 to 24 kg seeds kg^−1^ N_f_ for K uptake of 178 and 504 kg ha^−1^ and seed yield of 3.0 and 5.14 t ha^−1^, respectively [93]. Potatoes are considered to be a K-sensitive crop, in terms of both yield and tuber quality [104]. Studies on the effect of the N × K interaction on tuber yield have clearly shown that an increased dose of K fertilizer can significantly increase N productivity. This is an important premise for reducing the N_f_ dose [105,106].

## 6. Phosphorus

As in the case of K, the biophysical functions of P in plants are well-known and understood [60]. Three of its key functions can be treated as essential in agricultural practice. The first one concerns the biochemical energy (i.e., adenosine triphosphate; ATP), which is a high-P-energy compound. The synthesis of ATP is completely dependent on the supply of P to a plant. The generated energy is used in all plant energy transformation processes, starting with uptake of nutrients, and then their transport and transport of assimilates between the plant’s organs. At the end of this (energy and matter) transformation chain, the accumulation of organic compounds takes place in the main crop products (e.g., seeds, grains, roots, tubers, fruits) [107,108]. The second key function of P is the synthesis of nucleic acids (DNA and RNA), constituting 40–60% of the organic P pool in the plant. These compounds are components of genes and chromosomes, constituents of the plant’s genetic code. For this reason, they are responsible—among other aspects—for crop production; that is, the production of new generations of plants through seeds and grains [109]. The third crucial function of P, which is important in plant production, is phytin. This is a P storage compound in seeds/grains. The content of P in seeds is important, stimulating their germination and plant vitality in the early stages of growth [110,111].

Phosphorus is taken up by the plant from the soil solution as an orthophosphate ion (H_2_PO_4_^−^, Pi) [112]. The amount of P needed to maintain the optimum rate of plant growth is much smaller, compared to N and K [85]. In general, the P requirements of non-seed plants is much lower that of the seed- or fruit-producing plants, ranging from 14 to 40 kg P ha^−1^ [107]. As with nitrates, the uptake of orthophosphate ions is an energy-dependent process. The key reason is the high P concentration gradient of 10,000 between its concentration in the plant cell (cytosol) and the soil solution [113]. The uptake of P from the soil is, first of all, a function of the plant root density (RLD; cm roots cm^−3^ soil). The content of available P in the soil solution is the second factor determining its acquisition from the soil [114,115]. The mechanisms of P extraction from the soil are diverse, depending on the current P nutritional status in the plant. Plants deficient in P trigger a number of processes, such as investment in the RLD and root surface area, symbiotic associations with arbuscular mycorrhizal fungi, rhizosphere acidification, and activation of phosphate transporters in the plasma membrane. All of these processes undergo acceleration when Pi concentration in the soil solution decreases [116,117,118].

Phosphorus deficiency results from low Pi supply to the plant root, due to its low content in the soil solution or unfavorable environmental conditions, including low temperature, water shortage/soil drought, and low soil pH, among others [119]. The visual symptoms of P deficiency—regardless of the plant species—can be seen on older leaves as bluish–violet discoloration, leading frequently to plant death due to disturbances to basic physiological processes (Figure 6a). These symptoms, appearing in the early stages of plant growth, are an indirect signal of a deep disturbance in N metabolism [120]. Crop responses to P deficiency manifest as growth inhibition, even leading to a failure in development of reproductive organs, which is often observed in maize (Figure 6b). A mild deficiency causes a temporary, short-term slowdown in plant growth [121]. The first plant response to a slight P deficiency is very specific, being manifested by the ingrowth of roots at the expense of the shoot biomass [108,122]. In a P_i_-rich growth milieu, plants take up excess P to their needs, accumulating it an inorganic form. This pool is used during the growth of reproductive organs, such as seeds, grains, and tubers/roots [109,123].

Phosphorus yield-forming functions are best recognized for seed crops for which two critical stages have been identified. The first, minor function refers to all plants, and appears in the early stages of plant growth, being an important factor influencing the growth rate of roots and shoots [111,118]. For this reason, in plant production, a starting dose of P fertilizer is suggested to be applied, regardless of the content available P in the soil [124]. The second critical period for P uptake by dicotyledonous crops is poorly understood. In root and tuber crops, the increase in P accumulation is associated with the stage of intensive sugar or starch accumulation (Figure 7; [123,125,126]). As has been reported by Barłóg et al. [126], ¼ of the recommended P rate for sugar beets is sufficient to achieve a moderate yield (≈60 t ha^−1^ FW); however, in order to exploit the sugar beet yield potential in Poland (≈80 t ha^−1^ FW of storage roots)—which significantly depends on weather—the full P rate is needed. The key reason for this is that the storage yield depends on the supply of P to the plant during the late stages of growth. The advantage of a lower P dose (as shown in Figure 4) is the cessation of N accumulation in storage roots, resulting in both earlier sugar beet technological maturity and a lower content of so-called harmful N compounds [126].

In the case of seed plants, the main period of P requirement begins at the onset of flowering (Figure 3; [71,94]). Most of the P accumulated in the vegetative parts of these plants is then stored in seeds/grains. In a mature seed crop, between 85–95% of the total accumulated P is in seeds/grains [127,128]. The P remobilization efficiency ranges from 60–85% [129]. The degree of depletion of P in vegetative plant parts by the growing seeds/grain is not limited by its content in those organs, as has been suggested by Veneklass et al. [109]. In fact, it depends on the number of seeds/grains per plant that act as a physiological sink [127]. The key reason for the excess of P in vegetative plant parts is not the low rate of P remobilization from vegetative plant parts, as suggested by Wang et al., [130], but the low requirements of the growing seeds/grains [127].

For low-yielding seed plants, and such a model dominates in the world, the P plant resources accumulated before flowering are sufficient to ensure even a moderate yield. The essence of the matter is the dilution effect that P is subject to in the reproductive organs of the plant [127,131,132]. However, a completely different model of the P management functions in high-yielding seed plants. In the first stage of reproductive organs growth, P is effectively re-mobilized and then re-translocated from the vegetative plant parts to the growing seeds/grains. The P resources accumulated in the plant before flowering, as a rule, are sufficient to cover the needs of the reproductive plant’s organs. One of the reasons for this is the high P plasticity in seeds/grains, expressed by a very high rate of its dilution. This phenomenon has been observed for maize, wheat, rice, and winter oil seed rape [108,127]. In the second stage of seed/grain growth, the increased demand for P can be covered by P taken up by the plant from the soil [108,133]. This strategy is especially important for rice, which takes up 40–70% of the total P from the soil during the grain-filling period [129].

The reliability of long-term experiments for the assessment of P management by plants is limited by too high doses of P fertilizers used annually [65,66,67]. As a result, the recovery of P, even assessed over a long period, is typically low. As has been reported by Buczko et al. [66], the annual rate of P ranged from 10 to 210 (with an average of 60.7) kg ha^−1^ y^−1^. The relative increase in yield, even on soil naturally poor in the available P, did not exceed 10%. Consequently, these data cannot be used to determine the sufficiency level of available P (P-SL) in the soil for the tested plants. A high P recovery (P-R) is usually expected in soils poor in available P. As shown in Figure 8, wheat responded to P fertilization up to 45 kg P_2_O_5_ ha^−1^. This P dose increased the content of available P in the soil to a level sufficient for the maximum yield of wheat in this particular study. There was no increase in the yield above the range of 16–18 mg P kg^−1^ soil. Above this range, there was no response of wheat to an increase in the content of available P just before flowering [134]. In the presented example, the P recovery was 65% in the variant with 15 kg P_2_O_5_ ha^−1^, 50% in the variant with 45 kg P_2_O_5_ ha^−1^, and only 16% in the variant fertilized with 150 kg P_2_O_5_ ha^−1^. Moreover, in P-R, starting from a plot of 45 kg P_2_O_5_ ha^−1^, this perfectly fits into the power function:(8)$$P-R=1751P_{f}{}^{-0.94}{\;\mathrm{for}\;R}^{2}=0.999\;\mathrm{and}\;n=8$$

This example clearly demonstrates that P_f_ rates above that required for P–SL do not affect crop yield. Excessive P content in vegetative wheat parts, above P–SL, indicates insufficient size of wheat sink (the number of grain per unit area) to utilize these resources. This means that, for current wheat varieties, it is necessary to prepare new plant nutrition standards. Those that were developed 40–50 years ago are not reliable in current agricultural practice [135]. In the presented case, however, the most important fact is that the partial factor productivity of N_f_ (PFP-P_f_) increased from 28 to 45 kg grain kg N_f_^−1^ in the control P plot and 45 kg P_2_O_5_ ha^−1^, thus determining the appropriate P-SL range.

Two conclusions summarize this section: first, the shortage of P significantly reduces, while its excess supply does not affect, N productivity; second, the P_f_ rate should adjusted to the soil level of availability that maximizes the productivity of N_f_ of the currently grown plant.

## 7. Efficient Nitrogen Management—The Soil Fertility Clock Concept

### 7.1. Definition of the Concept

The role of N_f_ in crop production, which is the absolute basis of food production, results from its effects on the rate of plant growth, partitioning of assimilates between plant parts, formation of yield components, and quality of plant products [136]. The dominant impact of N on the life cycle of crop plants, as presented above, is justified at three levels of organization of the plant production process: (i) biochemical, (ii) physiological, and (iii) agronomic.

The key productive function of all other nutrients is to support the actions of N and, in fact, to control its productivity. The nitrogen use efficiency (NUE) is defined as the amount of main product per unit of N taken up by the plant during its growing season [137]. For this reason, any sophisticated attempt to calculate efficiency indices for K, P, or even for other nutrients is useless for a farmer. The calculation is useful, but only for understanding basic biophysical processes in crop plants [85,138,139]. The supply of N-SNs to the currently grown plant should be maintained at the level that maximizes—and not reduces—the yield-forming effects of N. It is necessary to take into account that the farmer cultivates plants in a specific sequence, called crop rotation. Therefore, the key goal of the farmer is to determine the critical level of soil fertility, not for all crops in rotation, but for the most sensitive plant to the nutrient, decisive for N productivity. This is the key challenge for the farmer, which should be treated as a necessary condition aimed at the development of an effective N_f_ management system for both farm economics and without imposing negative pressure on the health of the environment.

### 7.2. Maximum Attainable Yield—A Farm Production Goal

The potential yield of the grown plant—that is, the yield achieved under optimal environmental conditions (climate + soil) and rational management of the applied resources—is a theoretical term. This term, however, defines the production target, which may not be achievable in real production conditions [140,141]. Despite this, farmers need data on the maximum yields of crops that are grown in the geographical area of their production activity. This forms the basis for assessing the distance from the actual yield (Y_a_) to the realistic production target, namely, the maximum attainable yield (Y_attmax_). There are several ways to calculate or forecast Y_attmax_ and Y_a_. There is no doubt that the most important factor determining N uptake from the soil solution is water, the carrier of ions arriving at the root surface [60]. Therefore, one of the most frequently used methods for Y_a_ determination relies on water productivity, in terms of water use efficiency (WUE). This method assumes a fixed amount of yield per unit of water [142]:(9)$$\mathrm{WUE}=\frac{Y_{a}}{{\mathrm{ET}}_{a}}$$
where Y_a_ is the actual yield (kg, t ha^−1^) and ET_a_ is the water use (i.e., water transpired by a plant or evaporated from the bare soil; mm, m^3^);

This method assumes that a plant’s yield increases with an increasing amount of available water. Hence, the yield defined in this way is called “the water-limited yield” (WLY) [142]. This method has been modified by Grzebisz et al. [82], as follows:(10)$$\mathrm{WLY}=\mathrm{TE}\;\left(R\;-\;\Sigma E_{s}\right)+\mathrm{WR}$$
where TE is the transpiration efficiency (TE = *k*/VPD; *k* is the biomass transformation ratio), VDP is the vapor pressure deficit (hPa), R is the total sum of rainfall during the growing period of the cultivated crop (mm, m^3^), E_s_ is the seasonal soil evaporation (equal to 110 mm), and WR denotes the water reserves in the rooted soil zone (mm, m^3^).

The main component of this equation is TE. Its value for wheat in Australia has been estimated at 20 kg grain mm^−1^ of water, with a maximum of 30 kg grain mm^−1^ [143]. However, this index depends on the amount of water available during the growing season; for example, as reported for spring triticale grown in a humid climate (Poland), the TE value ranged from 15 to 39 kg grain mm^−1^ [93]. This wide range clearly indicates the high sensitivity of this index to soil fertility and, consequently, to the effect of nutritional factors on plant growth and yield. Moreover, WLY cannot be regarded as a constant climatic value, as it largely depends on the amount of available water during the growing season for the currently cultivated crop. As shown in Figure 9, for maize, the WLY ranged from 8.52 t ha^−1^ in a year with a normal pattern of weather conditions to 5.68 t ha^−1^ in a year with drought. The obtained yields, despite the completely different course of weather, were relatively high. The main reason for this was soil fertility (sandy loam, naturally rich in available K and other nutrients). The main conclusions that can be drawn from this Figure are as follows:(1)The content of available K (in the medium range) was enough to achieve the highest grain yield.(2)The interaction of K and N was observed, regardless of the course of the weather.(3)If the available K in the soil (being in the high range) is excessive, the yield will decrease.

The yield drop was accelerated by the increased dose of N_f_. This unexpected effect (i.e., yield suppression) was due to the excessive accumulation of N in maize biomass before flowering, which resulted in a reduction in the number of kernels per cob [144,145].

The analysis of the partial factor productivity of the N_f_ index (PFP-N_f_), for the presented case, is even more interesting. In 2001, the highest value of the index (109 kg grain kg N_f_^−1^) was recorded on a plot with high K level and fertilized with 100 kg N ha^−1^. The highest yield, however, was achieved on a plot with a medium K availability range and with 140 kg N ha^−1^, resulting in PFP-N_f_ of 100 kg grain kg N_f_^−1^. Both values are high, compared to the literature data [146]. In the presented case, it is worth discussing the rational choice of both N and K fertilizing systems. Raising the K soil fertility level to the high class resulted in a reduction in the dose of N_f_ by 40 kg ha^−1^. It is necessary to take into account the fact that, on the field with the medium K level, the application of 140 kg N ha^−1^ increased the yield by 29%, when compared to the treatment with 100 kg N ha^−1^.

**Figure 9 plants-11-02841-f009:**
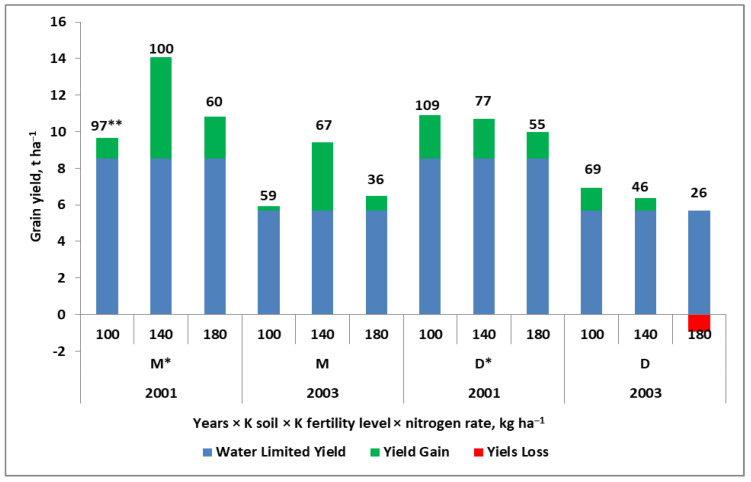
Effect of K soil fertility on maize yield in two years differing in water regime (modification based on [144]). Legend: * soil K fertility level: M, medium, D, high; ** Partial factor productivity of N_f_ (kg grain kg^−1^ N_f_).

The WLY concept is a good tool for scientific studies. In agricultural practice, its use requires a set of data that is not typically readily available to the farmer. Moreover, this method does not explain the action of factors responsible for WUE. A proposed alternative method for determining Y_a_ is the concept of nitrogen gap (NG) [91]. The main components of this methodological approach for yield gap calculation are the N_f_ applied to the crop (which is known to the farmer) and the main yield. This data set may be enriched with other environmental and agronomic data that impact the NUE. The calculation procedure consists of the following steps:(11)$$\mathrm{Partial}\;\mathrm{Factor}\;\mathrm{Productivity}\;\mathrm{of}\;N_{f}\text{: }{\mathrm{PFP}}_{\mathrm{Nf}}=\frac{Y}{N_{f}}\left({\mathrm{kg}\;\mathrm{kg}}^{-1}{\;N}_{f}\right)$$
(12)$$\mathrm{Attainable}\;\mathrm{maximum}\;\mathrm{yield}:\;\;\;\;\;\;\;Y_{\mathrm{attmax}}=c{\mathrm{PFP}}_{\mathrm{Nf}}\;\cdot {\;N}_{f}\left(t,\;\mathrm{kg}\;{\mathrm{ha}}^{-1}\right)$$
(13)$$\mathrm{Yield}\;\mathrm{Gap}:\;\;\;\;\;\;\;\;\;\;\;\;\;\;\;\;\;\mathrm{YG}={\;Y}_{\mathrm{attmax}}-{\;Y}_{a}\;\left({t\;\mathrm{ha}}^{-1}\right)$$
(14)$$\mathrm{Nitrogen}\;\mathrm{Gap}\;(N_{\mathrm{uw}}):\;\;\;\;\;\;\;\;\;\;\;\;\;\mathrm{NG}=\frac{\mathrm{YG}}{{c\mathrm{PFP}}_{\mathrm{Nf}}}\left({\mathrm{kg}\;N\;\mathrm{ha}}^{-1}\right)$$
where

N_f_ is the amount of applied fertilizer N (kg ha^−1^);

PFP-N_f_ is the partial factor productivity of N_f_ (kg grain/seeds, tubers, etc., per kg N_f_);

Y_attmax_ is the maximum attainable yield (t ha^−1^);

*c*PFP-N_f_ is the average of the third quartile (Q3) set of PFP_Nf_ indices, arranged in ascending order (kg grain/seeds, tubers, etc., per kg N_f_);

YG is the yield gap (t ha^−1^);

NG is the nitrogen gap (kg N ha^−^).

The advantage of this method over the WLY is its simplicity in determining both Y_a_ and Y_attmax_ in a well-defined soil–climatic geographical area. The farmer, having access to data on environmental and agronomic production conditions (e.g., soil texture, soil pH, contents of basic nutrients, plant variety, level of plant protection, date of sowing) in their production region, can determine factors limiting the Y_attmax_. The Y_a_ is a function of the formula:(15)$$Y_{a}=Y_{\mathrm{attmax}}-\mathrm{YG}$$

The NG values are required to produce a Y_attmax_ diagram, showing the distance between Y_a_ and Y_attmax_. The detailed procedure for preparing the NG diagram has been described in recently published papers [147,148]. The distance Y_a_ for a specific field from Y_attmax_ can also be expressed by the fractional Y_a_ index (Y_af_):(16)$$Y_{\mathrm{af}}=\frac{Y_{a}}{Y_{\mathrm{attmax}}}$$

A value of Y_af_ approaching 1.0 indicates sustainable management of applied N_f_.

### 7.3. Factors Affecting N Fertilizer Use Efficiency

As shown in Equations 1 and 2, the Y_a_ of the cultivated plant is the result of the interaction between Y_attmax_ and efficiency of applied N_f_ (EN_f_). In turn, the EN_f_ depends on the effectiveness of other factors that limit or increase the productivity of N_f_.

The total number of factors which impact EN_f_ can be divided into five main groups:(1)Farm organization and management;(2)Agronomic factors (e.g., cropping sequence ≈ crop rotation, soil tillage, seed bed preparation, cultivar, sowing date, harvest date);(3)Plant protection treatments, preventing yield reduction due to pathogens and pests;(4)Fertilizing treatments aimed at the correction of soil fertility;(5)Fertilizing treatments aimed at the in-season correction of the nutritional status of the grown plant.

The first group of production factors (i.e., organization and management of plant production processes on the farm) should not have a negative impact on the yield. In an economically well-run farm, the effectiveness of this group of factors should be at the level of 1.0—this is, after all, a basic prerequisite of the food production approach, known as the Sustainable Intensification of Agriculture (SIA) [14]. However, its implementation is only apparently easy. In fact, plant production on farms is under deep economic pressure [149]. A classic example is the method of plant cultivation. The economical decision to grow crops in monoculture, instead of crop rotation (a classical example is maize), may significantly reduce production costs, but a farmer must be aware of yield decreases [150]. There remains, however, uncertainty regarding the effectiveness of other production factors, which usually deteriorate. The decrease in yields of classic cereals grown under long-term monoculture is substantial. As shown in Figure 1, for winter rye, it can reach −20% under NPK treatment. The shortage of K resulted in a yield drop by 24% and that of P by 33%. There are also negative environmental effects to such an approach [151].

The farmer must effectively manage all agronomic factors and the health of the plant [61]. The efficiency of these factors, in accordance to the SIA concept, should be also set at 1.0. In this group, tillage and crop rotation are of particular importance for the available N pool and the uptake of its inorganic forms (N_min_) by plants [152,153]. The main task of soil tillage is to mix plant residues, manure, and mineral fertilizers—especially those containing low-mobility nutrients (P, K)—into the topsoil. No less important is the loosening of deeper soil layers and elimination of the plow sole [154]. The main production goal of this set of agronomic treatments is to increase the potential of the currently grown plant to penetrate the subsoil with its roots. It cannot be considered as only a source of water, as it is an important source of both N_min_ and N-SNs. Under conditions unfavorable to plant growth, such as drought, the yield is largely determined by water and nutrient resources in the subsoil [155,156,157]. Unfortunately, the greatest weakness of the current methods for diagnosis of soil fertility and the resulting fertilization recommendations (apart from N) is a lack of methods for assessing the capacity of these resources and the availability of N-SNs. A simple technical and diagnostic solution is the use of extractants for N_min_ [158,159].

The succession of plants grown on a given field is highly important for the effective management of N. A classic, biologically documented pattern of plant succession is the Norfolk rotation, which has been known for about two centuries [54]. As shown in Figure, 1, the cultivation of winter rye, a crop considered by farmers to be tolerant to monoculture, resulted in the significant yield reduction, which was exacerbated by the lack of balance of N with other nutrients. The cultivation of winter wheat (WW) after cereals also leads to yield reduction [67,160]. As has been reported by Babulicova [160], the yield of WW following legume plant was 29% higher than that following cereals. The well-established crop sequence is based on the assumption that dicotyledonous and monocotyledonous crops should be cultivated alternately in successive years. The advantages of crop rotation, regarding the efficient use of N_f_ are [65,87,161,162]:(1)The use of natural sources of N available on the field and farm. This leads to a reduction in the need for N_f_.(2)Biological subsoil amelioration by the strong root systems of dicots. Expected agronomic effects lead to:Mobilization of the soil nutrient resources (root exudates, mycorrhiza);Increased soil water capacity, resulting in better infiltration of rainwater;Increased exploration of the subsoil (i.e., growth of cereal roots in the root pores of dicots).(3)A narrower C:N ratio in manure and residues of legumes. Both sources of organic matter have a positive effect on humus formation and content in the soil.(4)The exploitation of soil nutrient resources within and in the soil profile is more sustainable, both qualitatively and quantitatively.

Crop rotation should not be considered by the farmer as a factor that decreases yield. Unfortunately, this is not the case, as has been evidenced in the scientific literature and agricultural practice. The key reasons for yield reduction due to wrongly planned crop succession (or even monoculture), to a great extent, are:Insufficient N supply during the critical stages of yield formation, reducing the main components of the yield [67];Disturbances in the processes of N transformation and uptake from soil [150];Disturbances in the uptake of nutrients responsible for NUE (Figure 1; [163]);Attack by pathogens, reducing the photosynthetic potential of the plant [164].

## 8. An Efficient System for Management of N-SNs—Principles of the Soil Fertility Clock

### 8.1. State of K and P Fertility Level and Food Production

The crop production potential of a single field in inherently related to its soil fertility level, presented as the depth of the humus profile and the content of available nutrients [165]. Fertile soil creates conditions for the build-up of a large (extensive) root system, which is crucial for the uptake of non-mobile nutrients, such as phosphorus and potassium [166].

The share of N, P, and K fertilizers in total nutrient uptake by cereals has been assessed as 33% for N, 16% for P, and 19% for K [167]. In China, as reported by Ren et al. [168], K fertilizer covers less than 20% (18.5%) of the total K in winter oilseed rape at harvest. Khan et al. [169], who studied 1400 field trials fertilized with K, did not observe any significant impact of the applied K (as KCl) on the yield of basic crops. According to MacDonald et al. [170], 29% of the world area of arable soils shows a deficit, while the remaining part shows a surplus of available P. The significant impact of P fertilizer can be revealed, as a rule, on soils with a low content of available P. The yield loss due to deficiency of P supply (P yield gap) to wheat is 22% (18–28%), 55% for maize (47–66%), and 26% for rice (18–46%). Moreover, the application of P fertilizers reduced this production gap to only 17% for wheat, 46% for maize, and 15% for rice [116]. At this point, it is necessary to ask whether the yield gap is actually due to a deficiency of available P, or the ineffectiveness of N_f_ due to the imbalance of P and other nutrients.

The key question to be formulated is: what is the appropriate level—or rather, the critical range—of N-SNs content in the soil? In the light of the facts presented above, classic P and K management strategies require significant modifications. The basis for these required corrections is the fact that the plants are grown in a cropping sequence determined by the economic goals of the farm. Not all modern crop sequences follow rational principles (i.e., biologically based crop plant succession) [54,171,172,173]. It has been well-documented, in millions of scientific articles, that the deviation of a particular cropping sequence from biological rules leads to a decrease in yield. The classic example is the Rothamsted long-term experiment with winter wheat [67]. Wheat followed directly by wheat or grown in monoculture yielded a significantly lower level, compared to that following dicotyledonous plants. Second, the maximum grain yield achieved under non-optimal rotation was, in this example, both lower and, at the same time, required a higher N_f_ rate. These results clearly indicate lower unit N_f_ productivity due to N immobilization, disturbance in uptake of N-SNs, and stronger pressure by pathogens [150,174,175,176].

The soil resources of P and K are the main source of nutrients for the cultivated crop. Over-exploitation of available pools of these nutrients leads to the degradation of soil fertility, subsequently resulting in the lower N_f_ productivity. Moreover, these processes create a multi-level risk, for the yield, farm economics, and the environment (Photos 2 and 3; [177]). The productivity of a particular field is determined by the level of soil fertility, conditioned by water capacity and the content of available N-SNs in the rooting zone of the currently cultivated crop [153,154,157]. At present, the subsoil resources of N-SNs are not included in typical soil fertility status diagnostic procedures. Moreover, there is a frequently presented opinion regarding their low significance for the plant growth and yield [178]. In the light of the published data, the share of P and K from fertilizers for crop plant nutrition is of secondary importance for the maintenance and synchronization of plant needs and nutrient supply from the soil. The logical conclusion to be drawn from these facts is unambiguous: the farmer’s goal for effective management of N_f_ is not to use P and K fertilizers or other carriers of these nutrients for direct feeding of the plant, but to coordinate the soil fertility level to maximize N_f_ use efficiency.

### 8.2. Management of Soil Fertility—Oriented to Cropping Sequence

Soil productivity is the ability of arable soil to provide the currently grown plant with air, water, and nutrients in the required amounts, mutual proportions, and ratios, ensuring full expression and development of the yield components [179]. This general property of arable soils has been indicated as one of the most important objectives listed in the Sustainable Development Goals (SDGs) by the United Nations in the 2030 Agenda for Sustainable Development [180]. This global goal can be achieved, but only through two targeted actions. The first is oriented towards stopping the degradation of soil fertility. This action refers to the world regions where soil fertility has been drastically reduced [10,177]. The second requires a significant correction in N-SN management strategies in areas of the world with advanced crop plant productivity. The so-called Old Agricultural Areas of the world will be decisive for food supply to the growing human population in the coming decades.

At present, two main concepts dominate in soil fertility management. The first—called the maintenance approach—is based on the assumption that the main goal of N-SNs is necessary to maintain the content of available nutrients at a certain level, allowing crop growth and yield. Three phases of soil fertility build-up can be distinguished using this approach: (i) build-up, (ii) maintenance, and (iii) draw-down [181]. In practice, the recommended rates of nutrients increase with the size of the gap between the maintenance level and the actual soil fertility status for a given nutrient. In most countries, using this fertilization approach, the nutrient doses recommended by agrochemical testing laboratories are consistent with the state of its deficiency. A classic example is the K recommendation in China for WOSR yielding at 3.75 t ha^−1^ [168]. The decreasing content of available K (NH_4_OAc-K extraction method) resulted in increasing the dose of applied K from 232 kg ha^−1^ at low K range to 50 kg ha^−1^ at high K range. The second approach, called sufficiency ranges, is based on the required (i.e., sufficient) level of the given nutrient for the respective crop [181].

These two fertilization strategies are based on the assumption that low-mobility nutrients are as effective as nitrate nitrogen [147]. The coefficient of effective diffusion for the N form is 2.7 × 10^−1^ cm^2^ s^−1^. In comparison, this index for K^+^ and NH_4_^+^ ions are about 100-fold lower (1–28 × 10^−8^ and 6.1 × 10^−8^ cm^2^ s^−1^, respectively). For H_2_PO_4_^−^ ions, this index is 10,000-fold lower, compared to nitrates [182,183]. The differences in the uptake rates of N and K are shown for sugar beets in Figure 7. Within 7 days of sugar beet growth in July, the N-NO_3_ resources, but not K, were completely depleted (100%) to a depth of 1.8 m. For K, this level of depletion was reached at a depth of 0.5 m. The most intensive uptake of both nutrients took place in the soil layer (0.0–0.6 m). There are two main conclusions to be drawn from Figure 10:(1)The nutrients are exploited from the whole root zone of the currently grown crop plant;(2)The faster uptake of nitrates than K^+^ by the fibrous roots of sugar beet means that a significant part of the low-mobility nutrients in the soil will not be taken up (i.e., not used up in the growing season).

**Figure 10 plants-11-02841-f010:**
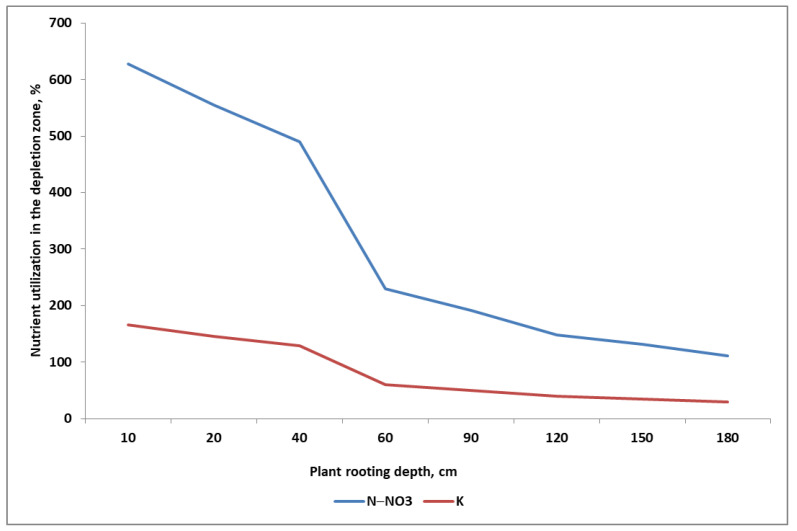
Degree of nitrate and potassium utilization in the soil during the maximum stage of K accumulation by sugar beets (based on [74]).

The classic concepts of N-SNs do not take into account two crucial facts; that crop plants differ in their sensitivity to the supply of N-SNs:During the growing season;In the course of crop rotation.

The efficient management of N-SNs in a soil/plant system should be based on the following principles of crop production:(1)Annual crop plants should be cultivated in a fixed sequence (i.e., the crop rotation).(2)Cereals have, as a rule, lower requirements for K, but higher requirements for P, compared to non-cereal crops.(3)Dicotyledonous plants have higher requirements for K than cereals.(4)The architecture of the root system of cereals is, as a rule, extensive compared to dicotyledonous plants. Consequently, cereals are less sensitive to the level of P and K fertility.(5)The distribution of low-mobility nutrients varies with depth.(6)Plants during the growing season differ significantly in the critical stages of nutrient requirement:Seed crops show critical periods, in terms of P requirements, during the vegetative (minor one) and reproductive (main one) periods of growth;All crops are sensitive to K during the linear phase of growth.(7)The critical period for N requirements by a seed crop is related to stages of seed/grain density formation.(8)The key yield-forming function of P in all crops is to accelerate the early rate of the plant growth.(9)The exploitation of P resources by seed crops, accumulated in vegetative parts before flowering, depends on seed/grain density.(10)The yield-forming function of K in
Seed crops is to strengthen N action;Dicotyledonous crops is acceleration of the early rate of the plant growth (mostly up to the rosette stage).

All of these points should be taken into account by the farmer during the process of development of the fertilization system.

Soil Fertility Clock (SFC) is an approach based on three assumptions which are key to the effective management of N_f_:(1)Critical soil fertility is the value or range of a soil nutrient’s content that is sufficient to provide it in the appropriate amount to the plant most sensitive to its supply in a given crop rotation.(2)Other, non-sensitive plants in the given crop rotation create the necessary time-frame for recovery of its original critical content.(3)The content of a specific nutrient cannot be a limiting factor in N uptake and utilization for any crop grown. Its fractional use efficiency, regardless of the actual plant in crop rotation, is ≤1.0.

The SFC concept is visualized in graphical abstract and explained in Figure 11 and Table 1. The critical K level for oilseed rape or any other dicotyledonous crop creates favorable conditions for the succeeding crop; that is, cereals, and most often wheat. The K level will still be high enough to cover the K requirement of wheat. A third crop in a certain cropping sequence—for example, maize—requires the farmer’s attention to adjust the K content. This is necessary only if the K content drops below the medium level. This is very probable when harvest residues are removed from the field (Table 1). The critical period of K correction, which must be oriented toward the so-called crop rotation critical K level is, in the discussed case, the spring barley growing season. This is a key term in the agronomic clock for determining both the level of K in the soil and determining the fertilization needs for the plant sequence spring barley → winter oil-seed rape. Managing P in crop rotation is a bit more complicated. It requires, regardless of the grown crop, the use of a starting dose of P fertilizer. Basic P fertilization complies with the principles presented for K. An additional component of an effective system regarding the full set of nutrients is foliar fertilization. This method allows for the correction of plant nutritional status—in fact, N action—but only at stages preceding the critical stages of yield formation.

## 9. Conclusions

Many factors directly and indirectly affect the yield of plants grown on the farm. The dominant one is N, which determines the dynamics of plant growth, partitioning of assimilates between the plant’s organs, the expression of yield components, and consequently the yield. Currently, effective plant production is based on the use of N_f_. The first step in an efficient management of N_f_ is to determine the maximum attainable yield of crops grown on the farm. This yield category is defined by the basic environmental factors, i.e., climate and soil fertility, but at the same time it is strongly modified by agronomic factors (crop rotation, soil tillage, plant protection). However, the most important of these factors is the N_f_ dose. The target yield can be achieved as long as the efficiency of N_f_ approaches 1.0, but at the same time the N_f_, regardless of the dose, does not reduce the yield. The synchronization of N demand, which varies in the plant’s life cycle with the rate of its uptake (in fact, nitrate) from the soil, is not dependent on its soil resources. This condition is met, but only when it is balanced with the supply of other nutrients (nitrogen-supporting nutrients; N-SNs). This, it can assumed that effective control of N_f_ efficiency does not only depend on its applied dose. The yield of high-yielding crops determines the interaction efficiency of N × P and N × K. The requirements of crop plants for P and K during the growing season are time-separated. The period of intensive biomass growth is a critical stage in the sensitivity of the plant to the supply of K. In seed plants, the N × K interaction predefines the development of yield components. The sensitivity of plants to the supply of P is revealed both in the early stages of growth and in the phase of yield realization. The second phase is crucial for seed plants. Phosphorus deficiency results in poor development of fruits, grains and seeds. A deficiency of both nutrients in the soil during the critical stages of yield formation results in both a decreased N_f_ efficiency, and consequently, a lower yield. The basic production unit on a farm is the field, on which plants are grown in a specific time-sequence, known as crop rotation. The condition for achieving the required level of N_f_ (≤1.0) efficiency is the high effectiveness of other production factors, which is to be set at ≤1.0. The operational basis of an effective control of N_f_ efficiency is the content of P and K in the soil, which should be oriented to cover the requirements of the most sensitive plant in a well-defined crop rotation. Thus, the main goal of P and K application to the soil is to restore their content in the topsoil to the level required by the most sensitive crop in a given crop rotation. The other crops grown in this cropping sequence provide the time-frame to actively control the distance between the current P and K content from the required critical ranges.

## Figures and Tables

**Figure 1 plants-11-02841-f001:**
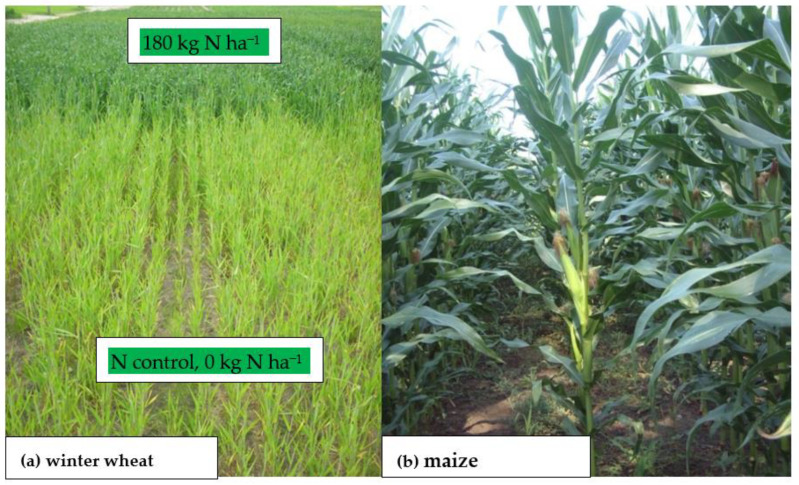
Impact of nitrogen fertilization on nutritional status of winter wheat and maize (photos by W. Grzebisz).

**Figure 2 plants-11-02841-f002:**
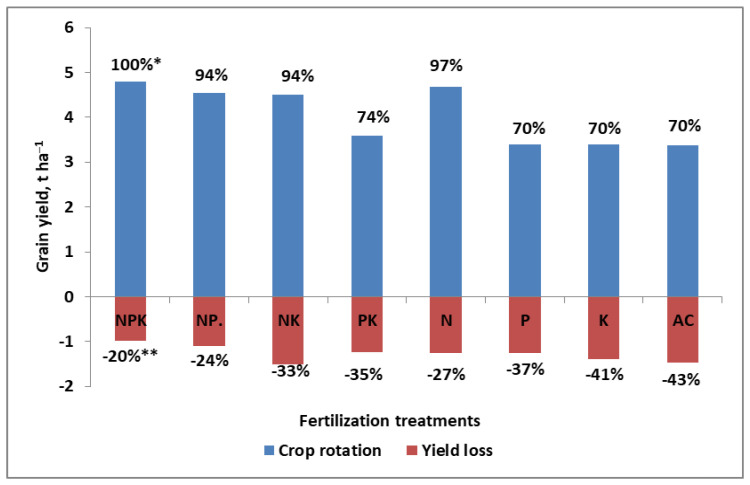
Effect of fertilization variants on yield of winter rye grown in 7-course crop rotation and monoculture (based on [65]). Legend: * Yield on the NPK variant = 100%; ** yield reduction—yield gap due to crop monoculture.

**Figure 4 plants-11-02841-f004:**
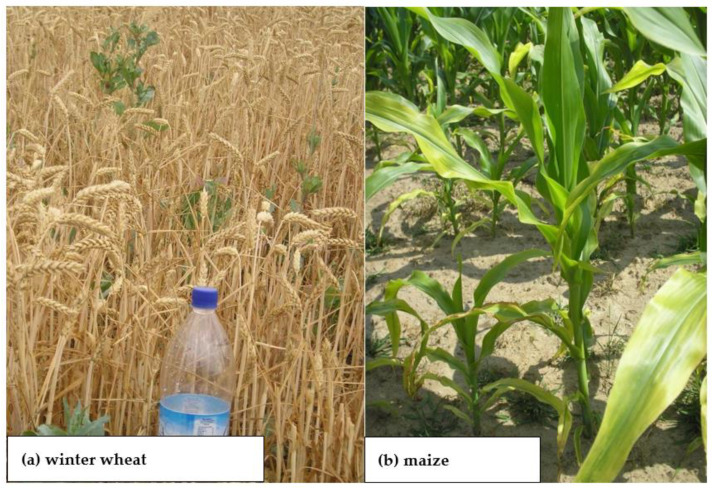
The stunted stature of plants due to potassium deficiency is the primary signal of yield depression in crop plants. (photos by W. Grzebisz).

**Figure 5 plants-11-02841-f005:**
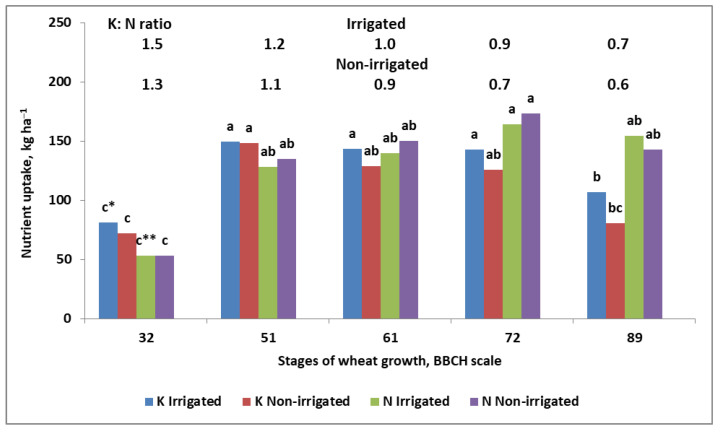
Nitrogen (N) and potassium (K) accumulation by winter wheat in critical stages of yield formation under various water conditions; means of three growing seasons (Grzebisz, not published). Legend: The different letters indicate significant differences between the treatments (*p* ≤ 0.05); * irrigation and ** non-irrigation conditions of wheat cultivation, respectively; K, N, potassium and nitrogen, respectively.

**Figure 6 plants-11-02841-f006:**
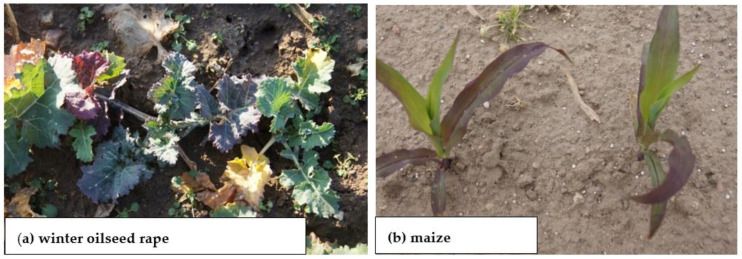
Symptoms of phosphorus deficiency in two different species grown in humid climate zone: (**a**) violet plants of winter oilseed rape are not able to conduct photosynthesis); and (**b**) violet maize plants at BBCH 33 often fail to develop a cob. (Photos by W. Grzebisz.)

**Figure 7 plants-11-02841-f007:**
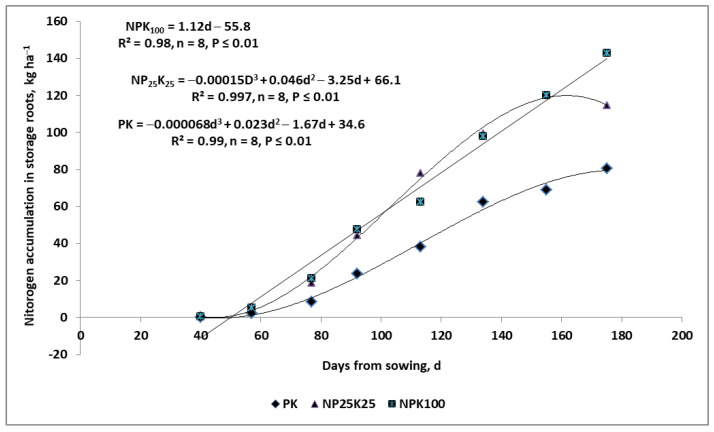
Effect of NPK fertilization systems on nitrogen accumulation in storage roots of sugar beets (based on Szczepaniak et al. [125]). Legend: PK—NPK100—the applied amount of P and K; N control; NP25K25—P and K applied at a dose of 25% compared to NPK100.

**Figure 8 plants-11-02841-f008:**
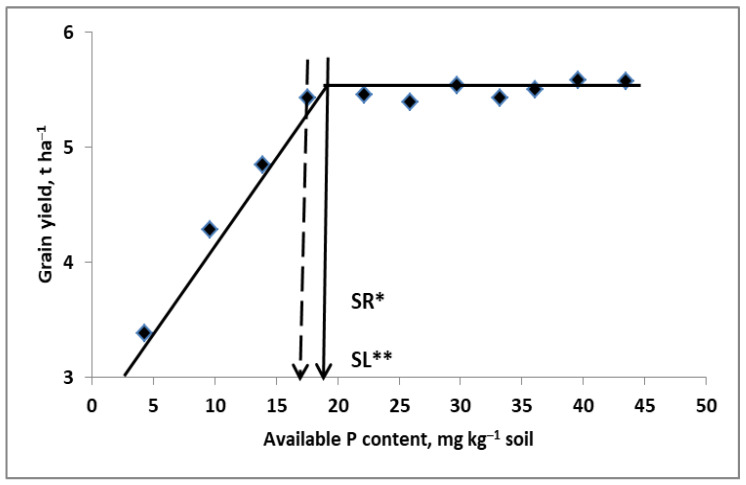
Yield of wheat response to the content of available phosphorus (based on [134]). Legend: * SR, ** SL, phosphorus sufficiency range, sufficiency level of the content of available P (determined by Olsen method).

**Figure 11 plants-11-02841-f011:**
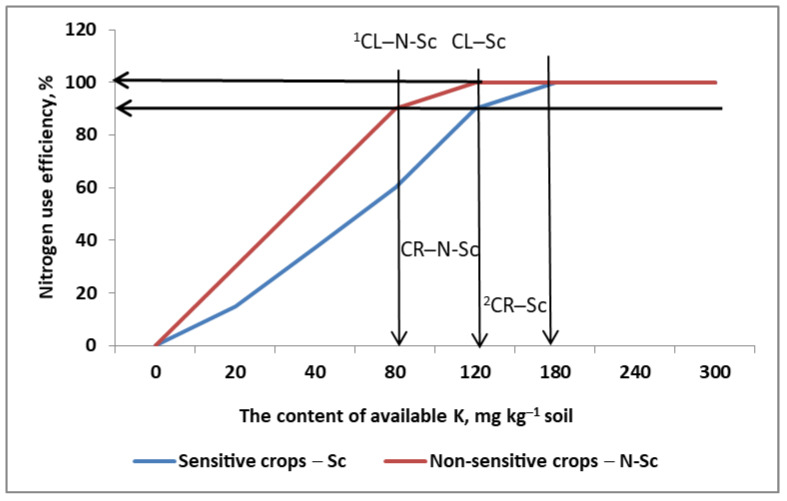
The crop rotation conceptual approach for the sufficient K level and range on sandy loam for sensitive and non-sensitive K plants. Legend: ^1^ CL–N-Sc, CL–Sc, CR–N-Sc, ^2^ CR–Sc: critical level and critical range for non-sensitive and sensitive plants to the content of available K (determined by the Egner–Riehm method).

**Table 1 plants-11-02841-t001:** Phosphorus and potassium balance in four-course rotation with winter oilseed rape. under full use of straw ^1,^*, t ha^−1^.

Crop Rotation	Nutrients			
	Phosphorus, P_2_O_5_		Potassium K_2_O	
	Demand	Full Recycling	Demand	Full Recycling
Winter oilseed rape, 3.5 t ha^−1^				
-seeds	70	-	35	-
-straw	40	20 **	200	180 **
Winter wheat, 7.0 t ha^−1^				
-grain	55	-	35	-
-straw	25	12	120	110
Maize, 8.0 t ha^−1^				
-grain	60	-	40	-
-straw	30	15	160	145
Spring barley, 5.0 t ha^−1^				
-grain	45	-	25	-
-straw	10	5	110	100
Total sum	335	53	725	535
Balance	−282		−190	
Total demand	**282**		**190**	
Partial demand, kg ha^−1^ year^−1^	**70.5**		**47.5**	

^1^ Simulation based on authors own data; * average range of soil fertility with P and K; ** recovery of P and K from straw during four-course rotation, phosphorus −50%, potassium −90%.

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
