# Peer review of "Soil Fertility Clock—Crop Rotation as a Paradigm in Nitrogen Fertilizer Productivity Control"

_plants, 2022, doi:10.3390/plants11212841_

Round 1

Reviewer 1 Report

The article presents previously unpublished data on the concept of Soil Fertility Clock (SFC) in which the authors present the assumption and found the basis on the critical content (range) of essential nutrients in the soil that has been adapted. By taking different nutrients and crops into consideration authors made the basis to the requirements of the most sensitive plant in the cropping sequence. The overall research and the findings given by the authors are good and could be published in this journal after some minor corrections and modifications.

Some comments and recommendations are listed below:

Please improve the quality of English grammar in the text, as well as check punctuations, spaces, incomplete sentences, spellings, etc.

Keywords should be properly mentioned.

Authors should check the MS for plagiarism as it was observed in line 45-49, 33% similarity, rephrase and rewrite the sentences. Also check the MS throughout for the punctuations.

In photo 1, authors mentioned in the caption ‘Impact of nitrogen on nutritional status of a. winter wheat and b. maize’, Is it possible to get an idea about the nutritional status of these crops by observing only photos? Explain.

In photo1, the control should be separated from the experimental as it will makes confusion to the readers. So, provide the separate images for the control and experimental.

Line 244. Check for punctuations.

Line 286-287. The legends should be mentioned in the figure, and explanation given in the figure caption not as separate, as it is mixed with the MS text makes confusion.

Line 306. Define WOSR? And then use its abbreviation.

Line 360 should be in the figure caption photo 2.

In Figure 3, Line 415-416, The legends should be mentioned in the figure, and explanation given in the figure caption not as separate, as it is mixed with the MS text makes confusion. Should be checked for spellings as well.

Line 502-503, The legends should be mentioned in the figure, and explanation given in the figure caption not as separate, as it is mixed with the MS text makes confusion.

Line 533, What does it mean ‘the P plant resources..’? sentence is unclear.

Line 576-577, The legends should be mentioned in the figure, and explanation given in the figure caption not as separate, as it is mixed with the MS text makes confusion.

Conclusion section is missing.

References should be checked for spacing, italics, bold, and proper formatting as per journal guidelines.

Author Response

Review Report 1_response

Please improve the quality of English grammar in the text, as well as check punctuations, spaces, incomplete sentences, spellings, etc.

The manuscript was checked by MDPI AuthorServices before being sent to the PLANTS Editorial Board. I can’t do more than specialists in this field. I would like to ask the PLANTS Editorial Board to act in this regard.

The certificate is attached (English-Editing-Certificate-50713.pdf).

Keywords should be properly mentioned.

The order of the keywords has been improved.

Authors should check the MS for plagiarism as it was observed in line 45-49, 33% similarity, rephrase and rewrite the sentences. Also check the MS throughout for the punctuations.

There is no plagiarism in this text, including this passage. This is an original conceptual work, based mainly on my group own research results and general considerations in this field. In order to avoid any associations, a slight verbal correction of this passage was made.

In photo 1, authors mentioned in the caption ‘Impact of nitrogen on nutritional status of a. winter wheat and b. maize’, Is it possible to get an idea about the nutritional status of these crops by observing only photos? Explain.

 These photos are condensation of nitrogen actions in crop plants production. Detailed explanation is in the text:

“The effect of N supply to the plant manifests itself in clear, visible changes in the architecture of the plant’s canopy. As can be seen in Figure 1a, wheat plants grown on an N control plot (i.e., without Nf supply) presented stunted growth (i.e., dwarf stature), low weight and surface area of leaves, and pale green color. In contrast, plants well-fed with N were characterized by a well-developed shoot, high mass and surface area of leaves, and an intense green color. All of these plants, despite a significant difference in the architecture of shoots, were in the same phase of growth (i.e., booting; BBCH 40-49). This phase is the crucial for the development of yield structure and determines the number of fertile florets [51]. Excess N supply to the plant, as shown for maize, results in the establishment of more cobs per plant (Figure 1b). However, this does not mean a higher yield of grain. Excessive supply of N has also resulted in excessive growth of non-productive plant parts, leading to a reduction in grain per unit area [52,53].”

In photo1, the control should be separated from the experimental as it will makes confusion to the readers. So, provide the separate images for the control and experimental.

 Appropriate explanations. Indicating the N fertilizer doses, are provided in Figure 1a. Detailed explanations are provided in the text (previous point).

Line 244. Check for punctuations.

 The manuscript was checked by MDPI Author Services before being sent to the PLANTS Editorial Board. I can’t more than specialists in this field. I would like to ask the PLANTS Editorial Board to act in this regard.

The certificate is attached (English-Editing-Certificate-50713.pdf).

Line 286-287. The legends should be mentioned in the figure, and explanation given in the figure caption not as separate, as it is mixed with the MS text makes confusion.

 The legend is explained directly in the figure. The captions under all figures have been separated from the MS text.

Line 306. Define WOSR? And then use its abbreviation.

 It has been explained in the text.

Line 360 should be in the figure caption photo 2.

 It has been corrected.

In Figure 3, Line 415-416, The legends should be mentioned in the figure, and explanation given in the figure caption not as separate, as it is mixed with the MS text makes confusion. Should be checked for spellings as well.

  The legend is explained directly in the figure. The captions under all figures have been separated from the MS text.

Line 502-503, The legends should be mentioned in the figure, and explanation given in the figure caption not as separate, as it is mixed with the MS text makes confusion.

  The legend is explained directly in the figure. The captions under all figures have been separated from the MS text.

Line 533, What does it mean ‘the P plant resources..’? sentence is unclear.

 It has been corrected to be self-explained.  

„For low-yielding seed plants, and such a model dominates in the world, the P resources accumulated in the plant before flowering are sufficient to ensure even a moderate yield. The essence of the matter is the dilution effect that P is subject to in the reproductive organs of the plant [128,132,133].”

Line 576-577, The legends should be mentioned in the figure, and explanation given in the figure caption not as separate, as it is mixed with the MS text makes confusion.

  The legend is explained directly in the figure. The captions under all figures have been separated from the MS text.

Conclusion section is missing.  The conclusion chapter has been added.

Conclusions

Many factors directly and indirectly affect the yield of plants grown on the farm. The dominant one is N, which determines the dynamics of plant growth, partitioning of assimilates between the plant’s organs, the expression of yield components, and consequently the yield. Currently, effective plant production is based on the use of Nf. The first step in an efficient management of Nf is to determine the maximum attainable yield of crops grown on the farm. This yield category is defined by the basic environmental factors, i.e. climate and soil fertility, but at the same time it is strongly modified by agronomic factors (crop rotation, soil tillage, plant protection). However, the most important of these factors is the Nf dose. The target yield can be achieved as long as the efficiency of Nf approaches 1.0, but at the same time the Nf, regardless the dose, does not reduce the yield. The synchronization of N demand, which varies in the plant’s life cycle, with the rate of its uptake (in fact, nitrate) from the soil, is not dependent on its soil resources. This condition is met, but only when it is balanced with the supply of other nutrients (nitrogen-supporting nutrients; N-SNs). This, it can assumed that effective control of Nf efficiency does not only depend on its applied dose. The yield of high-yielding crops determines the interaction efficiency of N × P and N × K. The requirements of crop plants for P and K during the growing season are time-separated. The period of intensive biomass growth is a critical stage in the sensitivity of the plant to the supply of K. In seed plants, the N × K interaction predefines the development of yield components. The sensitivity of plants to the supply of P is revealed both in the early stages of growth and in the phase of yield realization. The second phase is crucial for seed plants. Phosphorus deficiency results in poor development of fruits, grains and seeds. A deficiency of both nutrients in the soil during the critical stages of yield formation results in both a decreased Nf efficiency, and  consequently, a lower yield. The basic production unit on a farm is the field, on which plants are grown in a specific time-sequence, known as crop rotation. The condition for achieving the required level of Nf (≤ 1.0) efficiency is the high effectiveness of other production factors, which is to be set at ≤1.0. The operational basis of an effective control of Nf efficiency is the content of P and K in the soil, which should be oriented  to cover the requirements of the most sensitive plant in a well-defined crop rotation. Thus, the main goal of P and K  application to the soil is to restore their content in the topsoil to the level required by the most sensitive crop in a given crop rotation. The other crops grown in this cropping sequence provide the time-frame to actively control the distance between the current P and K content from the required critical ranges. 

References should be checked for spacing, italics, bold, and proper formatting as per journal guidelines.

It has been checked and corrected.

Witold Grzebisz

Reviewer 2 Report

The author explained literature very well. But before acceptance author revise following mistake

Add review aims at the end of abstract

Line 33; use full farm at first use “FAO”

Rewrite “reaching 67-, 42-, 38-, and 55%, respectively

Rewrite the legend of figure 1

Line 232, only use full farm at first use remove HI

LINE 242 “nitrogen use efficiency (NUE), only use abbreviation,

Line 246 remove from heading (N-SNs)

Avoid sentence start from abbreviation

Line 343 replace figure 2a

In the figure redraw and correct subscript in the figure such as K2O, P2O5

Replace the word photo with figure in the whole manuscript

Line 394 “replace with new words “In all years of study”

Some places author used the word photo and some places used figure.. ,,, only use figure

Line 558: rewrite “65-, 50-, and 16% on plots---

Line 661 and 661 combine with line 659 and 660.

Line 783 rewrite “assessed as 33-, 16-, and 19%, respectively.

Line 793; rewrite “to only 17-, 46-, and 15%, respectively.

Redraw the figure 7 and correct subscript in the figure such as N-NO3

LINE 903; Use full farm at first use.. “Soil Fertility Clock (SFC)

Line 935 to 938 combine into one paragraph

Please correct all grammatical errors and typos

Recheck all abbreviations.

Author Response

Review report 2 _ response

The author explained literature very well. But before acceptance author revise following mistake

Add review aims at the end of abstract

This is an original conceptual work, based mainly on my group own research results and general considerations in this field.

The aim of this manuscript is as follows:

“The Soil Fertility Clock (SFC) concept is based on the assumption that the critical content (range) of essential nutrients in the soil is adapted to the requirements of the most sensitive plant in the cropping sequence (CS).”

The key idea of The Soil Fertility Clock (SFC) concept has been reinforced by adding a graphic abstract.

Line 33; use full farm at first use “FAO”

It has been added.

Rewrite “reaching 67-, 42-, 38-, and 55%, respectively

It has been corrected.

Rewrite the legend of figure 1

It has been corrected.

Line 232, only use full farm at first use remove HI

It has been corrected.

LINE 242 “nitrogen use efficiency (NUE), only use abbreviation,

The double use „nitrogen use efficiency” (NUE) is intentional, so as not be confused with „nutrient use efficiency”.

Line 246 remove from heading (N-SNs)

It has been removed.

Avoid sentence start from abbreviation

 The manuscript was checked by MDPI Author Services before being sent to the PLANTS Editorial Board. I can’t do more than specialists in this field. I would like to ask the PLANTS Editorial Board to act in this regard.

The certificate is attached (English-Editing-Certificate-50713.pdf).

Line 343 replace figure 2a

In the figure redraw and correct subscript in the figure such as K2O, P2O5

It has been corrected, but I am unable to enter a subscript in Excel.

Replace the word photo with figure in the whole manuscript

It has been replaced in the whole manuscript.

Line 394 “replace with new words “In all years of study”

It has been replaced and sounds as follows”  

„The final K uptake, regardless of the seed yield and weather conditions during the research, always exceeded that of N.”

Some places author used the word photo and some places used figure.. ,,, only use figure

It has been unified.

Line 558: rewrite “65-, 50-, and 16% on plots---

It has been corrected.

Line 661 and 661 combine with line 659 and 660.

It has been combined.

Line 783 rewrite “assessed as 33-, 16-, and 19%, respectively.

It has been corrected.

Line 793; rewrite “to only 17-, 46-, and 15%, respectively.

It has been corrected.

Redraw the figure 7 and correct subscript in the figure such as N-NO3

The Figure 7 (now 11) has been corrected, but I am unable to enter a subscript in Excel.

LINE 903; Use full farm at first use.. “Soil Fertility Clock (SFC)

It has corrected.

Line 935 to 938 combine into one paragraph

It has been corrected.

Please correct all grammatical errors and typos

The manuscript was checked by MDPI Author Services before being sent to the PLANTS Editorial Board. I can’t more than specialists in this field. I would like to ask the PLANTS Editorial Board to act in this regard.

The certificate is attached (English-Editing-Certificate-50713.pdf).

Recheck all abbreviations.

All abbreviations have been checked.

Witold Grzebisz

Reviewer 3 Report

Comments to Author

This is a well written and time demanding manuscript. They briefly discuss about the importance and requirements of different plant nutrients for proper growth and development of plants. I have no questions about this manuscript.

However, I have some observation to publish this manuscript.

Ø  Author should add “Conclusion” section in the manuscript.

Ø  They should avoid the long sentences. Make the long sentences to simple sentences.

Ø  Follow the Author guidelines for ‘References’.

Ø  Added the “abbreviations” after Conflicts of Interest.

Author Response

Review report 3_response

This is a well written and time demanding manuscript. They briefly discuss about the importance and requirements of different plant nutrients for proper growth and development of plants. I have no questions about this manuscript.

However, I have some observation to publish this manuscript.

Ø  Author should add “Conclusion” section in the manuscript.

The conclusion section has been added.

Conclusions

Many factors directly and indirectly affect the yield of plants grown on the farm. The dominant one is N, which determines the dynamics of plant growth, partitioning of assimilates between the plant’s organs, the expression of yield components, and consequently the yield. Currently, effective plant production is based on the use of Nf. The first step in an efficient management of Nf is to determine the maximum attainable yield of crops grown on the farm. This yield category is defined by the basic environmental factors, i.e. climate and soil fertility, but at the same time it is strongly modified by agronomic factors (crop rotation, soil tillage, plant protection). However, the most important of these factors is the Nf dose. The target yield can be achieved as long as the efficiency of Nf approaches 1.0, but at the same time the Nf, regardless the dose, does not reduce the yield. The synchronization of N demand, which varies in the plant’s life cycle, with the rate of its uptake (in fact, nitrate) from the soil, is not dependent on its soil resources. This condition is met, but only when it is balanced with the supply of other nutrients (nitrogen-supporting nutrients; N-SNs). This, it can assumed that effective control of Nf efficiency does not only depend on its applied dose. The yield of high-yielding crops determines the interaction efficiency of N × P and N × K. The requirements of crop plants for P and K during the growing season are time-separated. The period of intensive biomass growth is a critical stage in the sensitivity of the plant to the supply of K. In seed plants, the N × K interaction predefines the development of yield components. The sensitivity of plants to the supply of P is revealed both in the early stages of growth and in the phase of yield realization. The second phase is crucial for seed plants. Phosphorus deficiency results in poor development of fruits, grains and seeds. A deficiency of both nutrients in the soil during the critical stages of yield formation results in both a decreased Nf efficiency, and  consequently, a lower yield. The basic production unit on a farm is the field, on which plants are grown in a specific time-sequence, known as crop rotation. The condition for achieving the required level of Nf (≤ 1.0) efficiency is the high effectiveness of other production factors, which is to be set at ≤1.0. The operational basis of an effective control of Nf efficiency is the content of P and K in the soil, which should be oriented  to cover the requirements of the most sensitive plant in a well-defined crop rotation. Thus, the main goal of P and K  application to the soil is to restore their content in the topsoil to the level required by the most sensitive crop in a given crop rotation. The other crops grown in this cropping sequence provide the time-frame to actively control the distance between the current P and K content from the required critical ranges. 

Ø  They should avoid the long sentences. Make the long sentences to simple sentences.

The manuscript was checked by MDPI Author Services before being sent to the PLANTS Editorial Board. I can’t do more than specialists in this field. I would like to ask the PLANTS Editorial Board to act in this regard.

The certificate is attached (English-Editing-Certificate-50713.pdf).

Ø  Follow the Author guidelines for ‘References’.

It has been checked and corrected.

Ø  Added the “abbreviations” after Conflicts of Interest.

It has been added.

Witold Grzebisz

Reviewer 4 Report

It is interesting to propose the concept of soil fertility clock in crop rotation for nitrogen management. Using the N as a critical indicator, the crop actual yield can be expressed by the maximum attainable yield with nitrogen efficiency, while nitrogen efficiency is determined by N-supporting nutrients. Such a paradigm is reasonable and acceptable. In this review, the authors interpret the relationship between nitrogen and the supporting nutrients mainly based on the amount and ratio, but without considering the using time. In practice, the fertilizer is used by 4R principles, namely, right source, right rate, right place and right time. Moreover, it is critical and a challenge to establish an effective Nf management system based on soil fertility and crop goal yield. The calculation of N-SNs is also a difficult mission for the crop in various CS. How to distinguish and understand the Equation 1 and 17?

Author Response

Review report 4_response

It is interesting to propose the concept of soil fertility clock in crop rotation for nitrogen management. Using the N as a critical indicator, the crop actual yield can be expressed by the maximum attainable yield with nitrogen efficiency, while nitrogen efficiency is determined by N-supporting nutrients. Such a paradigm is reasonable and acceptable. In this review, the authors interpret the relationship between nitrogen and the supporting nutrients mainly based on the amount and ratio, but without considering the using time. In practice, the fertilizer is used by 4R principles, namely, right source, right rate, right place and right time. Moreover, it is critical and a challenge to establish an effective Nf management system based on soil fertility and crop goal yield.

The calculation of N-SNs is also a difficult mission for the crop in various CS. How to distinguish and understand the Equation 1 and 17?

Equations 17 and 18, to avoid substantive confusion with equations 1 and 2, have been removed from the text.

The term of fertilizers application, more precisely the regulation of the level of soil fertility in P and K is presented by means of a graphic abstract.

The calculation of the fertilizer dose of P and K is basically the next stage in the SFC concept development. The precision in the dose of a given nutrient determining should also take into account its resources in the subsoil. Currently, it is a great challenge for agricultural sciences.

The manuscript was checked by MDPI Author Services before being sent to the PLANTS Editorial Board. I can’t do more than specialists in this field. I would like to ask the PLANTS Editorial Board to act in this regard.

The certificate is attached (English-Editing-Certificate-50713.pdf).

Witold Grzebisz

Round 2

Reviewer 2 Report

The authors have revised the manuscript according to my previous comments. Paper can be published now.

Reviewer 4 Report

The mansucript was improved and revised according to the comments. No further revision is needed.